# Recent Advances on the Design and Applications of Antimicrobial Nanomaterials

**DOI:** 10.3390/nano13172406

**Published:** 2023-08-24

**Authors:** Clara Ortega-Nieto, Noelia Losada-Garcia, Doina Prodan, Gabriel Furtos, Jose M. Palomo

**Affiliations:** 1Instituto de Catálisis y Petroleoquímica (ICP), CSIC, c/Marie Curie 2, 28049 Madrid, Spain; clara.ortega@csic.es (C.O.-N.); n.losada@csic.es (N.L.-G.); 2Department of Dental Composite Materials, Raluca Ripan Institute of Research in Chemistry, Babes-Bolyai University, 30 Fantanele St., 400294 Cluj-Napoca, Romania; doina_prodan@yahoo.com

**Keywords:** antimicrobial materials, nanomaterials, metal nanoparticles

## Abstract

Present worldwide difficulties in healthcare and the environment have motivated the investigation and research of novel materials in an effort to find novel techniques to address the current challenges and requirements. In particular, the use of nanomaterials has demonstrated a significant promise in the fight against bacterial infections and the problem of antibiotic resistance. Metal nanoparticles and carbon-based nanomaterials in particular have been highlighted for their exceptional abilities to inhibit many types of bacteria and pathogens. In order for these materials to be as effective as possible, synthetic techniques are crucial. Therefore, in this review article, we highlight some recent developments in the design and synthesis of various nanomaterials, including metal nanoparticles (e.g., Ag, Zn, or Cu), metal hybrid nanomaterials, and the synthesis of multi-metallic hybrid nanostructured materials. Following that, examples of these materials’ applications in antimicrobial performance targeted at eradicating multi-drug resistant bacteria, material protection such as microbiologically influenced corrosion (MIC), or additives in construction materials have been described.

## 1. Introduction

The current global challenges in industry, healthcare, and the environment have prompted the exploration and study of new materials in ways that can help combat major problems and needs through new approaches.

Within this development, nanomaterials have been a key part. Their applications are limitless and are present in fields as varied as environmental remediation [1]; the development of renewable energy sources or energy storage [2]; the improvement of beauty and health treatments [3,4]; or the production of improved building, protective, or coating materials [5,6,7] (Figure 1). In addition, the nanomaterials have also been shown great potential in combating bacterial infections and addressing the challenge of antibiotic resistance.

In particular, metal nanoparticles (MeNPs) have been described to show excellent characteristics to inhibit different type of bacteria and microorganisms [8,9,10].

The potential effect of nanoparticles in comparison with bulk material is because of the nanoscale size provides a large surface-area-to-volume ratio that maximises the contact area and reactivity of the material, resulting in increased in catalytic properties, including the antibacterial [11,12,13,14] and antiviral activity [15,16].

Furthermore, a diverse range of nanomaterials have been developed, considering polymeric materials, silica, earth rare elements, carbon-based materials such as carbon nanotubes, and graphene derivatives that have been employed with different properties themselves or directly doped by metal particles as hybrid materials [17]. For example, superhydrophobic coating materials have been recently developed with antimicrobial characteristics [18] or strategies of coating antimicrobial peptides as emerging promising dental implant coating material to reduce/prevent bacterial infections [19].

In this review article, we address some of these recent breakthroughs in nanomaterial preparation and their application as antimicrobial agents for a variety of industrial purposes.

## 2. Design and Synthesis of Different Nanomaterials

Nanoparticles can be synthesised directly, obtaining the nanoparticles alone, or structured in multiple and diverse ways, where they are combined with other compounds, metals, or materials.

Currently, there are two main approaches to nanoparticle synthesis: top-down and bottom-up. The first generally comprises physical synthesis methods, while the second usually consists of chemical and biological (also known as “green”) methods [1,20]. Some of the most common are chemical reduction, coprecipitation, seeding, microemulsion, sol–gel process, hydrothermal synthesis, electrodeposition, ball milling, laser ablation, sputtering, pyrolysis, or the use of plant or microorganisms, among others [3,21,22]. The disadvantage of some of them would lay in high energy consumption, expensive equipment, high pressure and/or temperature [18,23], and the pollution caused by the use of toxic reagents or solvents [24].

There is also a wide variety of ways to coat or functionalise their surface, as well as to generate multifunctional nanoparticles. Their combination in different nanostructures can improve their stability as well as certain properties both in the materials with which they are combined and in themselves, generating synergistic effects.

For example, an efficient way to conjugate nanoparticles is using polymers or biopolymers [25]; proteins [26,27,28,29]; or other materials such as silica [30,31], graphene [32,33,34], or plant extracts [35,36].

Here, we collect some of these variations, where NPs are synthesised in the form of nanostructures, combined with other metals or functionalised, with the aim of improving their antimicrobial activity, applied in different fields.

### 2.1. Synthesis of Metal Nanoparticles (Ag, Cu, Zn, Ce)

Silver nanoparticles (AgNPs) and their derivatives have been studied for decades. There are plenty methods to synthetise them, coming from very different approaches. AgNPs have superior characteristics compared with conventional silver, having been experimented upon in different fields such as medicine, energy, catalysis, or bioremediation [18,20,21,37] Various synthesis processes are suitable for the formation of AgNPs. Chemical reduction, the sol–gel method, photochemical techniques, or biosynthesis are effective methods to manufacture AgNPs and require only a metal precursor, a reducing agent, and a stabilising agent. They are simple to perform, but also controllable, leading to the acquisition of spherical nanoparticles [18,21,38,39]. The stabilising agents can be the reducing agents themselves, such as citrate of sodium [3]. Abdulsada and Hammood [7] used a hydroxyapatite/silver (HAp/Ag) coating on 2507 duplex stainless steel (DSS), placed by electrophoretic deposition (EPD), in order to improve corrosion and antibacterial resistance of DSS. Different deposition times (1 to 3 min) were tested to optimise the EPD parameters, and the voltage applied was 20 V. The experimental data showed that the coated substrates exhibited higher corrosion resistance than the uncoated material, producing a higher potential as the deposition time increased. In addition, SEM analysis exposed that as the deposition time increased, the amount of porosity in the coating decreased, being better than the distribution of particles at 3 min of EPD.

Copper is a metal that stands out for its antimicrobial capacity, abundance, and competitive price compared to others, such as Au or Ag. Thus, copper nanoparticles (CuNPs) have been the subject of studies. However, the disadvantages of CuNPs are their low stability under atmospheric conditions, which could lead to undesired oxidation, and the high cost and low sustainability of methods that generate them in pure form [40]. Various strategies have been used to avoid this problem: from obtaining them in an inert gas atmosphere [41], through applying protective polymers [25,42] or surfactants [43] to the use of biological synthesis methods, which are considered safer and ecological [44,45,46]. Moreover, in the case of copper, a key aspect is the control of the oxidation state and shape of the generated species, as this results in completely different properties.

A novel strategy for the synthesis of metal nanoparticles has been developed by the Palomo group ([26,27,47]), using an enzyme as a stabilising agent, generating metal–enzyme nanobiohybrids The enzyme facilitates the in situ formation of the nanoparticles under very mild synthesis conditions (water and room temperature), where the addition of reducing agents is not necessary. In addition, the use of a protein allows for the control of the size and shape of the metal nanoparticles produced, as well as the ability to produce them in a monodisperse way (Figure 2).

However, a recent study conducted by our group also demonstrated the effect of include a reduction step in the synthesis process of nanobiohybrids, where depending on the amount of reducing agent added, changes in metal structure are achieved as well as different antimicrobial efficiencies [48]. In the synthesis method, copper sulphate as the metal salt, lipase from *Candida antarctica* (CAL-B) as the scaffold enzyme, and sodium borohydride as the reducing agent were used. Different hybrids were synthetised using percentages of NaBH_4_ from 0% to 100%. XRD and TEM analyses demonstrated different oxidation states and sizes for the CuNPs depending on the reduction degree. With a higher amount of NaBH_4_, Cu(0) at a greater presence was found, as well as larger NPs, with a maximum size of 13 nm. With a lower amount of reducing agent, the predominant species were Cu(I), and the minimum particle size was less than 6 nm. In addition, an increase in the reducing agent generated larger and agglomerated nanobiohybrids (Figure 3).

We also can find processes such as the chemical and physical vapour deposition and different etching and pattering technologies, such as laser texturing (LT). Selvamani et al. [49] fabricated laser-textured copper (LT-Cu) with micro- and mesoporous structures by the scanning of a ND:YAG laser beam (in raster mode) with speed and power of 20% and 85%, respectively.

Laser texturing (LT) is a versatile and rapid processing approach to selectively melt and vaporise the surface of a targeted material to generate micron and nanoscale patterns without altering the bulk properties of the material. Selective laser texturing of the copper surface enhances the photothermal processing of the material. The in situ generation of mesoporous structures under ambient conditions made the process more scalable and cost-effective (Figure 4).

It is well known that metal surfaces with an oxide layer tend to show higher surface wettability [50,51]. The wetting properties of Cu and LT-Cu were characterised by water contact angle (WCA) measurement. As demonstrated in the study, the LT-Cu process creates a textured, highly rough surface covered by a hydrophilic metal oxide coating, resulting in superhydrophilic behaviour.

Zinc can be obtained at nanometric size in the form of zinc nanoparticles alone or zinc oxide nanoparticles [52]. However, nano zinc oxide (ZnO) is the most widely used zinc nanoparticle. ZnO is known to be a good semiconductor and to play an essential role in life. It also has low toxicity and is biodegradable. For this reason, and because of its low cost and simple preparation, it is used in food and cosmetic products [53].

In one study, researchers prepared silane-treated zinc oxide nanoflakes (S-ZnNFs) in which poly(methyl methacrylate) (PMMA) was incorporated to enhance the interfacial interactions in the nanocomposite (Figure 5). Zn nanoflakes are characterised by their 2D layer, which can enhance the mechanical properties of the structure. Their addition to PMMA is a good strategy in the development of new synthetic resins with mechanical and biological applications [54].

Furthermore, in recent years, the synthesis of semiconductors materials, in particular CeO_2_ NPs, has also received attention due to its diverse properties. CeO_2_ NPs can be prepared by wet-chemical synthesis, starting from cerium nitrate and oxalic acid, and applying a thermal treatment after the precipitation [55].

In another strategy, Algethami et al. [56] created CeO_2_−SnO_2_ nanofibers of 170 nm via the electrospinning method using a polyacrylonitrile polymer. The composite exhibited higher specific capacitance and improved photoelectrochemical performance, which may be attributed to the synergism between CeO_2_ and SnO_2_ particles in the nanofibers.

### 2.2. Synthesis of Carbon-Based Metal Hybrid Nanomaterials

Recent years have seen an exponential increase in the production of carbon-based nanomaterials such carbon nanotubes, graphene and its derivatives, nanodiamonds, fullerenes, and other nanosized carbon allotropes. Because of their small size, which is similar to that of many fundamental biomolecules; large specific surface area; high electrical and thermal conductivity; special optical properties; and superior mechanical properties, which have opened the door for a wide range of applications, carbon nanomaterials have an infinite number of potential modifications and tailoring [57,58].

In particular, graphene oxide (GO) is a versatile material with useful properties for many applications. It is extremely light, having a large specific surface area and a good electrical and thermal conductivity. Due to the functional groups with oxygen (hydroxy, epoxy, carboxy) present on the edges and on the surface of the graphene sheets, they can be easily synthesised by chemical oxidation and graphite exfoliation. In addition, graphene acts as a good vector for the stabilisation and immobilisation of metal nanoparticles and can be functionalised in a simple way with other different compounds or materials [59,60].

The use of graphene or other carbon materials, together with metallic nanoparticles, can lead to very interesting synergistic effects, as well as favouring the stability of the materials involved.

For example, Prodan et al. [60] developed and characterised powders of graphene oxide (GO) with silver (GO-Ag) and with zinc oxide (GO-ZnO), silanised with (3-aminopropyl) triethoxysilane (APTES). The functionalisation of GO reactive groups with organosilanes can improve the biocompatibility and other properties such as the bonding strength [61,62]. In addition, the presence of Zn and Ag metal nanoparticles may improve the antibacterial properties, being sought in this case the improvement of different characteristics of building materials.

Similarly, Han et al. [63] created a two-dimensional nanomaterial composed of GO, AgNPs, and sulfadiazine (a broad-spectrum antimicrobial agent) for biomedical applications. They introduced sulfadiazine and AgNPs onto the surface of GO to improve the antimicrobial activity of the hybrid material, taking advantage of the high specific surface area of GO. GO was previously functionalised with hydrophilic macromolecules to improve the aqueous stability of the hybrid material. In first place, as is shown in Figure 6, the GO surface was carboxylated with ClCH_2_COONa and functionalised with polyethylene glycol (PEG). Then, AgNO_3_ was introduced, and a microwave irradiation route was used to generate AgNPs. After that, sulfadiazine was added to obtain the hybrid nanomaterial.

It is also possible to incorporate metal NPs on the surface of carbon nanotubes, as Seo et al. did [64]. They combined CuNPs into multi-walled carbon nanotubes to improve the maximal antibacterial properties and minimal the cytotoxicity to human cells. This method allows the CuNPs to be kept stable and simultaneously minimise their oxidation. CuNPs were formed on the functional groups attached to the nanotube surface, previously treated in an acid mixture, by the NaBH_4_ reduction of copper ions from different salts, cupric chloride, and cuprous chloride. TEM, EDS, and XRD results demonstrated the production of monodispersed CuNPs distributed over the entire surface of the MWCNTs, with an average diameter of 7.05 nm and 9.22 nm for cuprous/MWCNTs and cupric/MWCNTs, respectively.

### 2.3. Multi-Metallic Hybrid Nanostructured Materials

Nanomaterials composed of a combination of different metals offer several advantages over single-metal nanoparticles. Key benefits include enhanced properties, resulting in synergistic effects and tailor-made features, multifunctionality, stability, or cost efficiency.

In terms of antimicrobial efficiency, hybrid nanoparticles also generate improvements. For example, the combination of Ag and CuO nanoparticles can lead to core–shell configurations with a synergetic enhanced antibacterial effect [65]. The doping of semiconductor metals as nickel or copper, with other metals as silver [66] or zinc [67], also improves the antibacterial activity, as well as the combined sputter deposition of silver with copper or platinum [16] or the combination of different metalloid-based antimicrobials, such as silver nitrate, copper sulphate, or nickel sulphate [68].

One approach is the functionalisation of metallic nanoparticles such as silver, copper, or zinc on other compounds, whether metallic or non-metallic. For example, nano silver species can be functionalised in aluminium hydroxide oxide (AlOOH) composites with nickel-doping (Ag-AlNiO), as Cheng et al. demonstrated [66], in order to generate a broad-spectrum antimicrobial activity against Gram-negative and -positive bacteria, as well as fungi. In the synthesis process, the first nickel-doped AlOOH nanoflowers were produced. After that, AgNPs were generated in the nanoflowers in a process involving solvents and temperature, as is shown in Figure 7. High-resolution TEM images evidenced the formation of a nanoflower-like morphology composed of corrugated nanosheets gathering abundant AgNPs in a uniform distribution with a size of 6-10 nm. In addition, the comparison with the composite without the Ni doping showed a weakly assembled effect of Ag on the AlOOH.

Polymers can be also used as stabilising agents in the synthesis of bimetallic nanohybrids, such as in the case of Maruthapandi et al. [67]. They created two polymers decorated with zinc-doped copper oxide (Zn@CuO) nanoparticles. For this purpose, they synthetised the compounds PANI-Zn@CuO and PPY-Zn@CuO, first forming polyaniline (PANI) or polypyrrole (PPY) by UV irradiation using carbon dots from their corresponding monomers, and then sonicating with copper acetate and zinc acetate. The PPY-Zn@CuO composite consisted of agglomerated spherical particles with diameters from 1 to 5 µm, while PANI-Zn@CuO showed irregular stick shapes with similar diameters (Figure 8). EDX and SEM-EDX analysis confirmed the presence of the metal oxide CuO.

In a different method, Meister et al. [16] developed ultrathin nanopatches of Cu-Ag and Pt-Ag, sputtering the metals on silicon substrates. They looked for synergetic effects through the combination of two different metals. Specifically, the improvement of the antimicrobial effectiveness of Ag^+1^ based on the electrochemical mechanism of the sacrificial anode through the combination with a nobler metal was sought.

Different variations of the process were carried out, as is shown in Figure 9. On the one hand, the sputtering time was varied to obtain layers of different thicknesses (60–630 s). On the other hand, the sputtering of the different metals was made homogeneously or sequentially, modifying the final nanostructures obtained.

SEM and TEM images revealed that thick films were continuous and showed different nanocrystalline surface microstructures. However, the structures in the nanopatches were discontinuous. In the case of sequential sputtering, the elements tended to be more separated as compared to co-sputtering, which tends to mix elements on the atomic scale and instead forms an alloy comparable to the co-existence of elemental films. Nanopatches were obtained using two different deposition times, 60 and 120 s. For 60 s of exposure, layers of 1.02, 2.4, and 2.4 nm were obtained for Pt, Ag, and Cu, respectively, while for 120 s of exposure, the layers were 2.04, 4.8, and 4.8 nm, respectively. These values were obtained independently of the type of deposition.

### 2.4. Superhydrophobic Coating Materials

Another type of system to obtain materials with antimicrobial properties is focused on the fabrication of superhydrophobic surfaces. The superhydrophobic effect is characterised by air pockets that form between the coating and planktonic bacteria in the water phase, greatly reducing the number of anchoring sites that are available for the bacteria to stick to and the likelihood that the bacteria will come into contact with the coating. To create a superhydrophobic coating, it is therefore necessary to combine a micro-/nanostructure with a surface modification using highly nonpolar ligands. In this vein, Akbulut and co-workers [18] developed a new type of coating system based on the controlled immobilisation of lysozyme into interstitial spaces of presintered, nanostructured thin film based on 200 nm silica nanoparticles. This was performed on aluminium sheets. Finally, the sequential chemisorption of an organofluorosilane to the available interfacial areas was performed. The material showed a contact angle of water of 159°, whereas base Al showed a contact angle of 73°, indicating a successful transformation from a hydrophilic to a superhydrophobic surface.

Another recent strategy was established using mussel-inspired adhesion capacity by catechol groups with silver nanoparticles in order to produce a superhydrophobic metallic material to combat microbiologically influenced corrosion [69]. Starting from basalt scales, an etched process takes place using concentrated NaOH solution for 2h, and then polydopamine (PDA) is deposited for 12 h and subsequently silver nitrate is added, producing AgNPs deposited on the surface by catechol oxidation. Finally, the introduction of 1-dodecanethiol by Ag-S interaction introduced the superhydrophobicity to the material. Superhydrophobic basalt scales were assembled onto epoxy resin matrixes on the surfaces of metal substrates to prepare composite coatings.

## 3. Antimicrobial Performance of Different Metal Combinations and Nanostructures

### 3.1. Metal Nanoparticles

In terms of antimicrobial efficiency, nanomaterials, depending on their structure or nature, show a different action mechanism, for example, depending on the metal used or functionalised material.

The antimicrobial properties of some metals, such as silver, copper, or zinc, have been known for centuries. However, in recent years, their use as hybrid materials, combined with others, in the form of nanoparticles or nanostructures, has broadened their spectrum of application as antibacterial and antiviral agents, as well as their efficacy [16].

One of the major advantages of using nanomaterials against microorganisms is that they act in a different way to traditional mechanisms. There are several mechanisms of action described for the antimicrobial performance of nanoparticles. We can differentiate between the action mechanism in the extracellular or intracellular environment of the cells. In the extracellular environment, nanoparticles can damage the cell membrane. This can occur through direct degradation of compounds that form the membrane, or indirectly through the generation of reactive oxygen species (ROS), which produce membrane damage [9,70] (Figure 10).

The most common ROS are peroxides, hydroxyl radicals, and singlet oxygen, and they can vary according to the metal involved [71,72]. In addition, some ROS can enter the cell by diffusion, while others such as OH– cannot penetrate and are placed on cell walls, disrupting cellular functions. Those that can penetrate, or those that are generated inside the cell, from the NPs which have entered by diffusion or by taking advantage of membrane imbalance, eventually disturb the ROS balance inside the cell and cause the destruction of DNA and proteins. This biomolecular damage ultimately induces the bacterial death and has a bactericidal role [53,70].

**Figure 10 nanomaterials-13-02406-f010:**
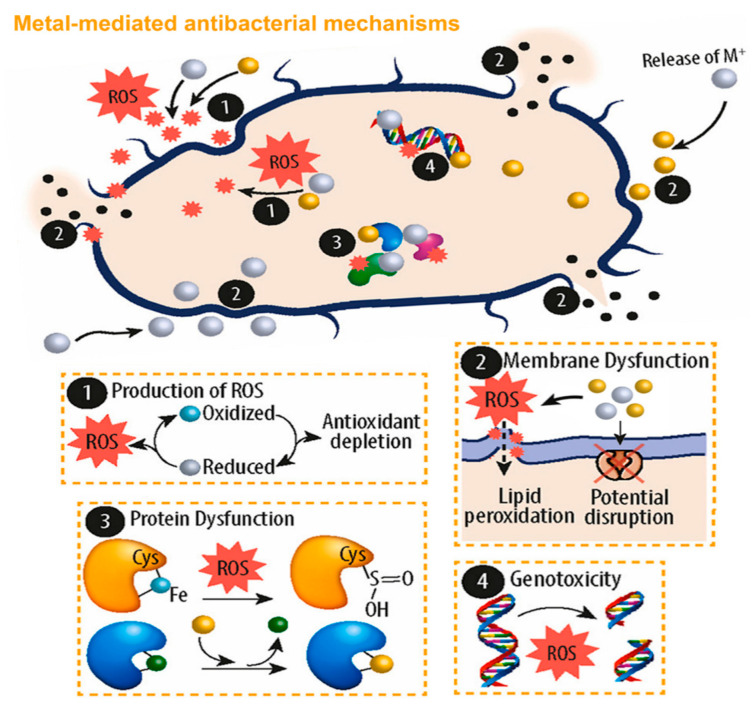
Scheme of antibacterial mechanisms of metal ions and NPs. The central modes of action are (1) formation of extracellular and intracellular reactive oxygen species (ROS); (2) interaction of ROS and metal nanoparticles with the cell wall; (3, 4) interaction with DNA and proteins. Adapted from [72].

Thus, the composition of the membrane of the target microorganism is a key factor in the effectiveness of nanomaterials. In addition, the microbicidal effect of metal NPs and ROS generation is also dependent on several factors; these include, but are not limited to, the physical form of NPs, the chemical state of metal (oxidation state), the concentration of NPs in contact with microbes, the proximity of microbes to surfaces that contain NPs, the form of application (wet or dry), the application temperature, and the presence of buffers or contaminants in the chemical environment [70].

### 3.2. Antimicrobial Mechanism of Carbon-Based Nanomaterials

Two main antibacterial mechanisms have been described for carbon-based nanomaterials (CBN): physical damage and oxidative stress (Figure 11).

Physical damage may be caused mechanically by the section of the cell membrane, due to the particular shape of the CBNs. It leads to the leakage of the intracellular matrix, and, finally, to bacterial death. For example, with graphene, the physical damage is produced by the “sharp” edges of graphene nanosheets, being consider as a “nano-knife” by some authors [73,74,75]. It has also been suggested that the larger CBN, as graphene oxide sheets, can envelop the bacteria, preventing their proliferation [76]. It is also possible to obtain the destabilisation of the bacterial membrane by electrostatic interaction with the nanomaterials [74].

On the other hand, the generation of oxidative stress is another important antibacterial mechanism of CBN. This can take place through a ROS-dependent or ROS-independent mechanism. The ROS-dependent mechanism acts as in the metal NPs described above. In the ROS-independent mechanism, the transfer of electrons is produced between the bacterial membrane and the CBN surface. The CBN would act as an electron acceptor, and it would be the surface the inducer of the antibacterial effect [73,76].

## 4. Applications

We here concentrate on investigating three of the numerous antimicrobial applications of the nanostructured materials presented above: their potential use as antibacterial agents in medical uses such as against multidrug-resistant (MDR) bacteria, as coatings for material protection or in surface disinfection, or as additives in building materials.

### 4.1. Medical Application: Nanomaterials against Multi-Drug-Resistant Bacteria

Multi-drug-resistant bacteria are a silent pandemic that deserves society’s attention. The development of new materials to combat them is currently a health priority, and nanomaterials are perfect candidates for this purpose, as they act against bacteria through different mechanisms than traditional drugs, as discussed above.

Cheng et al. designed and synthetised the multi-metallic nanostructure [66] described in Section 2.2. The antimicrobial power of Ag-AlNiO was tested with different MDR *Mycobacterium tuberculosis* strains and other Gram-negative and -positive bacteria, via microplate Alamar Blue assay at the concentrations of 2–256 μg/mL and disk-diffusion assay, respectively. The results indicated a superior antimicrobial activity of the doped composite compared with the non-doped one (Ag-AlO), with minimum MIC values of 16 and 32 μg/mL for *P. aeruginosa* and *M. tuberculosis* BCG, respectively. Disk-diffusion assays also revealed good antimicrobial properties against *E. coli*, *P. aeruginosa*, *S. aureus*, and *C. albicans* of both nanocomposites, but especially Ag-AlNiO, with larger inhibition zones.

It was suggested that the nickel-doped nanocomposite contributed to a regulated release of silver nanoparticles (Figure 12), which results in long-term antimicrobial activity. The data obtained also proposed that the nanocomposite Ag-AlNiO may be useful in reducing the spread of *M. tuberculosis*.

On the other hand, Maruthapandi et al. [67] synthesised and evaluated the antibacterial activities of two polymers decorated with zinc-doped copper oxide (Zn@CuO) nanoparticles (described in Section 2.2). The bacterial potency of these compounds was evaluated against *Escherichia coli* and *Staphylococcus aureus*. The bacterial activity was performed at the minimum composite concentration of 1 mg/mL, comprising 0.144 mg/mL of Zn@CuO. The results revealed a synergistic effect of these compounds compared to PANI and PPY and Zn@CuO. In the case of PANI-Zn@CuO, it completely eradicated *E. coli* after 24 h of exposure, while it eradicated *S. aureus* after 12 h. A noticeably shorter time of 8 h was observed when *E. coli* and *S. aureus* were exposed to the PPY-Zn@CuO compound. This difference was analysed using ROS species with an oxygen stress assay. The effect of oxygen stress was evaluated using the electron paramagnetic resonance (EPR) technique to measure ROS generation with DMPO (5,5-dimethyl-1-pyrroline-N-oxide) as a spin trap [77]. DMPO captures hydroxyl radicals (OH·) and superoxide anions (O^2-^) to create DMPO–OH as a final adduct, with a distinct quartet signal and a typical signal intensity ratio of 1:3:3:1.

The PPY polymer produced more ROS compared to its counterpart, PANI, but it was lower than the ROS level of Zn@CuO. Consequently, the ROS level of PPY-Zn@CuO was higher than that of PANI-Zn@CuO. The synergistic effect between PANI and PPY with Zn@CuO was attributed to the free electrons in the polymers, which could combine with the O_2_ in the suspension to enhance ROS production [78].

Another way in which copper nanoparticles can be used as antimicrobial agents is through their functionalisation with proteins, as mentioned above. In our study described in Section 2.1 [48], copper nanoparticles were synthetised with a lipase protein and different amounts of reducing agent to obtain the best antimicrobial performance against *Escherichia coli*, *Klebsiella pneumoniae*, and *Mycobacterium smegmatis*. The results revealed the differences in the antimicrobial efficacy of the different nanostructured materials. Variations with a higher degree of reducing agent were better against Gram-positive bacteria, while those with a lower degree were better against Gram-positive bacteria.

In a different approach, nanoparticles can be combined with different antibiotics in order to generate a synergistic effect against MDR bacteria. For example, Vorobyev et al. [79] combined silver nanoparticles with different cephalosporins, achieving better antibacterial activity with AgNP–antibiotic hybrids than with free AgNPs and antibiotics.

In a similar way, Han et al. [63] developed a nanomaterial of graphene oxide, AgNPs, and sulfadiazine, as explained above, and they tested the antibacterial activity of the material against *Escherichia coli* and *Staphylococcus aureus* (Figure 13).

The results demonstrated that the antibacterial activity of the hybrid nanomaterial was improved by over three times compared to the system without the antibiotic. The antibacterial performance of the hybrid was due to the triple synergistic effect of GO, AgNPs, and sulfadiazine: bacterial capping, puncture, and inhibition [63].

Superhydrophobic coating materials of nanosilica fluorocompounds on aluminium surfaces, previously described [18], showed excellent results, with the inhibition values of 6.5 (>99.99997%) log-cycle reductions in bacterial surface colonisation against Gram-negative *Salmonella typhimurium* LT2 compared to uncoated aluminium.

Cerium dioxide is a rare-earth metal, which makes it less well studied as an antimicrobial agent. Even so, various investigations demonstrated that CeO_2_ NPs have remarkable antibacterial activity. For example, Pop et al. [54] showed antibacterial activity of CeO_2_ NPs in relatively low concentrations against two Gram-positive and three Gram-negative bacteria by using the microdilution method. Also, the CeO_2_−SnO_2_ nanofibers prepared by Algethami et al. [55] demonstrated higher antimicrobial activity than simply CeO_2_ nanofibers when they were screened against *Escherichia coli*.

In summary, it has been shown that the use of nanostructured materials can be helpful in combating MDR bacteria, and that their efficacy is greater than other materials or simple nanoparticles. In addition, the use of low-cost materials and environmentally friendly synthesis methods are key factors in the development of new antibacterial compounds.

### 4.2. Material Protection Application: Nanomaterials as Coating Agents

An important utility of nanomaterials is in the protection of other materials of interest in the industrial or healthcare fields. For example, some applications include their use as coating agents to prevent biocorrosion generated by microorganisms in the maritime field or in pipelines, known as microbiologically influenced corrosion (MIC) [80,81]. Also, they are used to prevent microbial dissemination and proliferation, such as in fabrics [6,15,82], disinfectants [83], or biomaterials, like dental materials [84,85].

For this purpose, various synthesis techniques are applied, including direct surface synthesis or immobilisation, as explained previously. Other processes have also been applied where NPs are created directly on a surface, on which they are immobilised. Immobilisation of the nanostructures on a surface minimises their exposure to undesirable agents and their release into the environment, and even enhances their activity, as previously discussed.

In a novel process, described in Section 2.1, Selvamani et al. [49] developed laser-textured copper (LT-Cu) to enhance the antibacterial properties of copper. As an advantage, this method could be easily scaled up, allowing it to be used in industrial processes.

In the first place, contact-killing properties of Cu and LT-Cu were determined by placing a 7.1 log10 CFU mL^−1^ bacterial suspension of *E. coli* on both surfaces for 120 min. The results showed only a bacterial reduction of 1.5 log10 CFU mL^−1^ in samples exposed to a pristine copper surface. However, complete eradication of bacteria was observed in samples exposed to the LT-Cu surface. The superhydrophilic property and the higher Cu_2_O content on the LT-Cu surface are believed to be two main reasons for improving the effective contact between bacteria and the LT-Cu surface, which contributed to its superior antibacterial property [77,86].

After that, the bacterial kill profile was studied by immersing Cu and LT-Cu in 1 mL of PBS with 25 μL of bacterial strains, including two clinically relevant multidrug-resistant bacterial strains (*P. aeruginosa* and *MRSA USA300*) (Figure 14). Initially, 5 log10 CFU mL^−1^ of *P. aeruginosa* and *MRSA* were exposed to Cu and LT-Cu. Although Cu was able to eradicate *P. aeruginosa* and *MRSA* with 60 and 90 min exposure, LT-Cu was able to eradicate both pathogens at a relatively faster rate, with full eradication at 40 and 90 min, respectively. In addition, the bactericidal properties of LT-Cu and Cu were tested against larger inocula of *S. aureus* and *E. coli* (8 log10 CFU mL^−1^). LT-Cu was able to completely eradicate both bacterial strains in 120 and 60 min of exposure, respectively. However, the pristine Cu surface was unable to completely kill *S. aureus* and *E. coli* bacteria, even after 150 and 90 min of exposure.

LT-Cu was able to completely eradicate *S. aureus* in 120 min, twice the time required to eliminate *E. coli*. The long time required to kill Gram-positive bacteria (such as *S. aureus* and *MRSA*) can be linked to their thick peptidoglycan coating, which makes them rather more resistant to contact killing on copper surfaces [87]. The fivefold shorter overall time to eradicate Gram-positive pathogens strongly supports the superior bactericidal nature of the Cu_2_O mesoporous structure on the LT-Cu surface.

In another method explained previously, Abdulsada and Hammood [7] created a coating made up of hydroxyapatite and silver (HAp/Ag). A coating of HAp over biomaterials improves the resistance to corrosion, but also enables the growth of bacteria. To avoid this, they combined AgNPs with HAp by EPD in DSS to enhance the resistance of corrosion and the antibacterial resistance.

Antibacterial activity of HAp with and without AgNPs in suspension was studied using the disk diffusion method with *Escherichia coli* and *Staphylococcus aureus*. The results showed a significative decrease in the population of *E. coli* in HAp with AgNPs sample, compared to the control and the HAp alone. Regarding *S. aureus*, the results indicated that with the HAp alone, the growth was greatly inhibited, indicating a lower bacterial adherence in the presence of this material. Also, with the combination of HAp/AgNPs, more than 99% percent of bacteria growth was inhibited.

On the other hand, Meister et al. [16] developed ultrathin nanopatches of Cu-Ag/Pt-Ag (Section 2.2), with the objective of improving the antimicrobial properties of silver. Antibacterial tests were made with *Staphylococcus aureus*. The analysis of planktonic and adherent bacteria was performed in 24 h by the incubation of blood agar plates and fluorescence imaging, respectively. Different nanopatches were tested, and the best antimicrobial effect was found for the combination of Ag-Pt sequentially deposited, and Ag-Cu co- and sequentially deposited. However, no significant antimicrobial effect was found for single-metal nanopatches (made of pure Pt, Ag, and Cu), nor for co-deposited Ag-Pt (Figure 15).

In addition, antiviral tests were performed by the inoculation of SARS-CoV-2 onto the surfaces for either 1 h or 24 h. The best effect was found for Cu-Ag nanopatches, with a maximum reduction in viral titre of 3.9 log10 in 24 h for thick layers (120 s). In contrast, Ag-Pt nanopatches showed a mild effect (1 log10 reduction of viral titres) (Figure 15), and no effect was observed for Ag and Cu (thin layer-60s) pure nanopatches. Regarding the differences in the sputter deposition process, thin layers (60 s) reduced viral titres to a lesser extent than thick layers (120 s), but no differences were found in co- and sequential deposition. Thus, in the developed nanopatches, Ag-Cu combinations have higher antimicrobial activity than pure metals. A sufficient ion release is necessary to produce an antimicrobial effect, and possibly that generated by the pure nanostructures is not enough, as well as in Ag-Pt systems.

New nanostructured materials can also be very useful in the fight against bacterial biofilms in fields as diverse as dentistry, food processing, water systems, marine biofouling, and the oil and gas industry. Garcia et al. [84] investigated the antimicrobial effect of nano zinc oxide (ZnO) adhesives against saliva-derived biofilms. The nano ZnO was incorporated in different wt % in an experimental dental adhesive. First, a characterisation of the material was carried out, and after that, the antimicrobial activity was evaluated through the assessment of colony-forming units (CFU), metabolic activity, and live/dead staining (CFU counts of streptococci, mutant streptococci, and total microorganisms were performed).

The results showed that the adhesive with the highest percentage of ZnO (7.5%) led to a 50% reduction in the metabolic activity of the microorganisms in the biofilm and had lower CFU compared to the other candidates, specifically for total microorganisms. Microscopy images of live/dead assay showed more areas of dead microorganisms for the adhesives with higher ZnO percentages.

In another approach, Kim et al. [56] developed zinc oxide nanoflakes (S-ZnNF) on PMMA with the aim of preventing the adhesion of microorganisms and the biofilm formation due to its widespread use in dentistry and medicine. Thus, they enhanced the interfacial interactions in the nanocomposite in the presence or absence of UV treatment (Figure 16).

Li et al. [85] developed pH-responsive nanocomposites of AgNPs in combination with an antibiotic, kanamycin, and coated with polydopamine (PDA@Kana-AgNPs), with the aim of killing bacteria and disperse biofilms. Biofilms of *Staphylococcus aureus*, *Streptococcus pneumoniae*, *Pseudomonas aeruginosa*, and *Escherichia coli BL21* were treated with control groups and the nanocomposite for 6 h. Then, biofilms were stained with crystal violet for light microscopy observation and were quantified using the colorimetric assay.

After that, the first of them were still pyknotic and thick. However, after treatment with PDA@Kana-AgNPs, the biofilms became sparse and discrete, indicating their dispersion.

In addition, the antimicrobial effect of the nanocomposite was tested with the resazurin assay, which showed that the combination of kanamycin and AgNPs had greater antimicrobial activity against the tested strains than when applied separately (Figure 17).

Another interesting case is the superhydrophobic/AgNP-coated basalt scales, which had a combined effect of reducing bacterial adhesion to the surface and controlling Ag ion release, slowing down the deterioration of bacteria to epoxy resin coatings [69]. This material inhibited the growth of *Pseudomonas aeruginosa PAO1 bacteria* and the formation of their biofilms by the superhydrophobicity based on catechol and dodecanethiol groups and silver nanoparticles. The synergistic action of these two components exerts a more vital ability towards more corrosion resistance compared with those of surfaces employing a single strategy.

### 4.3. Building Materials Application: Nanomaterials as Additives

Nanotechnology can also be applied to building materials. For example, iron oxide can be used to colour bricks or other building materials by providing colour fastness [88]. The addition of nanosilica improves the mechanical properties and stability of concrete [89]. Several studies have evaluated the influence of nanomaterials on the properties of geopolymer-based mortars and concretes, both fresh and hardened, in different conditions [90,91]. Most of the reported data focused on the evaluation of the mechanical properties of building materials based on geopolymers, cured at ambient temperatures, being able to expand the obtained information in order to anticipate other geopolymer application [92]. However, the addition of Ag, Cu, ZnO, or GO nanoparticles in the building materials can also generate a potential antibacterial effect [91,93,94]. This may be beneficial to prevent microbial proliferation and the deterioration of surfaces in the construction and cultural heritage conservation fields [5,95,96].

In addition, there are reports in the literature about the addition of supplements with antimicrobial properties in the building materials (concrete, mortar, bricks, etc.), which do not significantly affect the essential properties of the materials [97,98,99].

Singh et al. [100] evaluated the antimicrobial effect of a cement–ZnO nanoneedle composite with different additions of ZnO (5%, 10%, and 15 wt.%), against *Escherichia coli*, *Bacillus subtilis*, and *Aspergillus niger.* It was observed that the antibacterial and antifungal effect of cement improved with the increasing ZnO concentration, being enhanced by the sunlight (Figure 18).

Furthermore, the cement–ZnO composites showed a significant improvement in all functionalities compared with the pure cement.

Metal nanostructured nanoparticles can also be incorporated into other building materials. In their study, Prodan et al. [60] used different graphene oxide (GO) powders functionalised with silver (GO-Ag) and with zinc oxide (GO-ZnO), which were explained in Section 2.1, using them as additives in hydraulic lime mortar composition.

They evaluated the antibacterial effect of non-silanised powders and the silanised ones against five bacterial strains (*Streptococcus mutans*, *Porphyromonas gingivalis*, *Enterococcus faecalis*, *Escherichia coli*, and *Staphylococcus aureus*) through the disk diffusion method. The results showed that the diameters of the inhibition zones ranged between 12 and 22 mm. The largest diameter of the inhibition zone was found for the GO-ZnO-APTES sample against *Enterococcus faecalis.* However, against the other strains, GO-Ag-APTES showed better antimicrobial activity. On the other hand, the antibacterial effect of the silanised and non-silanised samples depends on the tested bacterial strains. In conclusion, a satisfactory antibacterial effect was obtained for all functionalised, silanised, or non-silanised powders, and the inclusion of these powders as additives in hydraulic mortars can lead to an improvement in their quality.

Klapiszewska et al. [101] incorporated zinc oxide NPs with a cement matrix in order to achieve antibacterial properties in cement composites. They used different mixing techniques to evaluate the differences in the antimicrobial efficacy. It was a study from two different approaches: a direct contact method and optical density measurement. The results indicated that the method used to produce the final cement composite had a significant impact on its ability to inhibit microbial growth.

## 5. Conclusions

This review paper provides an overview of the manufacture and use of several nanostructured materials, many of which primarily contain metallic nanoparticles, in an effort to potentially produce an antibacterial effect. Various synthetic techniques, with a focus on environmentally friendly technologies that provide control over the metal species or shape of nanoparticles, are used. There are other intriguing cases where hybrid systems that combine metal nanoparticles with carbon-based materials have been described, illustrating a potential synergistic impact. The process by which the various materials work is one of the key ideas in antibacterial effectiveness. The possibility for various action mechanisms depending on the metal or substance, has been examined in this review. Finally, examples of how these materials may be used as antibacterial agents in various applications have been described, indicating how good and promising these kinds of materials are for protecting human health in the future. Combining new technologies to create various composite materials might have a future use in reducing the biocorrosion of materials in many infrastructural and industrial sectors, including water and wastewater treatment, transportation, and energy production. Future uses for these nanoparticles might include their capacity to be incorporated into environmentally friendly construction materials, biocidal materials for electronic devices, or even brand-new antibacterial textiles for use in hospitals.

## Figures and Tables

**Figure 1 nanomaterials-13-02406-f001:**
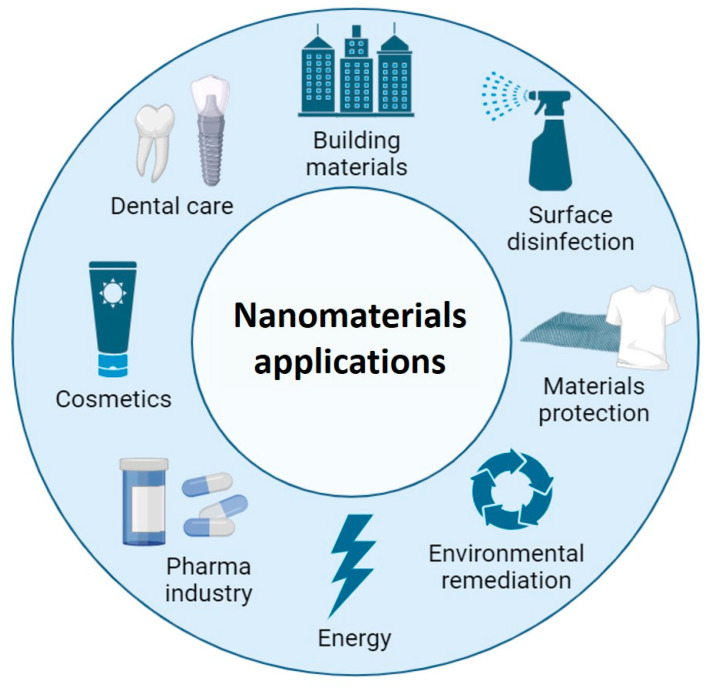
Potential antimicrobial applications of nanomaterials in different fields. Created with BioRender.com.

**Figure 2 nanomaterials-13-02406-f002:**
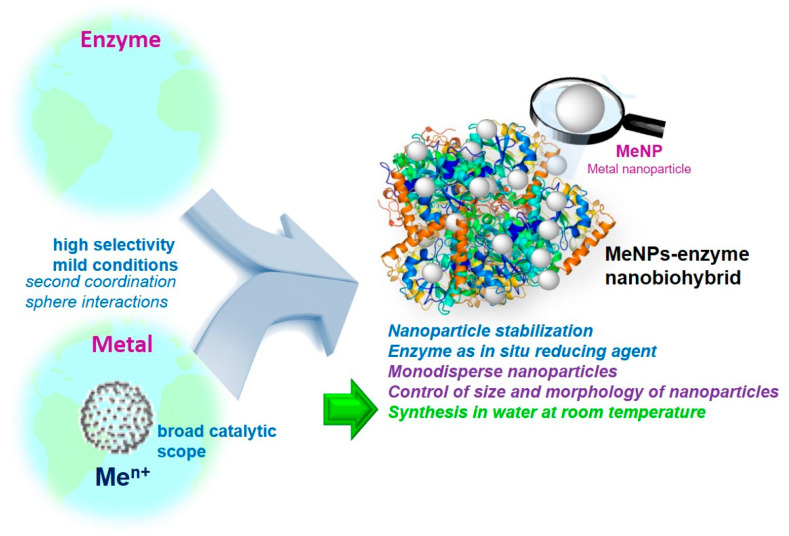
General concept and advantages of new enzyme–metal nanoparticle nanobiohybrids [26].

**Figure 3 nanomaterials-13-02406-f003:**
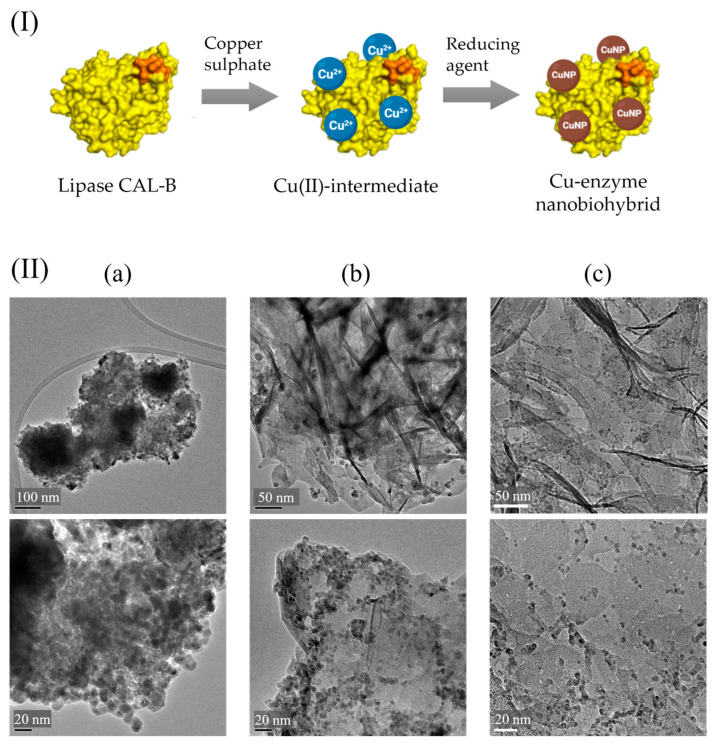
(**I**) Scheme of copper–enzyme nanobiohybrid synthesis. (**II**) TEM images of Cu–enzyme nanobiohybrids with (**a**) 100% of reducing agent; (**b**) 30% of reducing agent; (**c**) 10% of reducing agent. Adapted from [48].

**Figure 4 nanomaterials-13-02406-f004:**
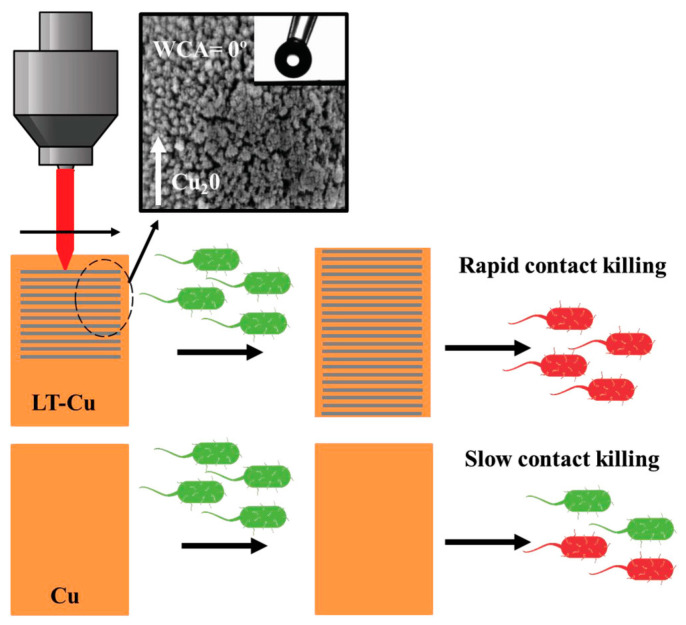
Representative scheme of efficient contact killing properties of laser-textured copper (LT-Cu). The one-step LT increased Cu_2_O percentage and induced the formation of superhydrophilic hierarchical micro/mesoporous structures, with a water contact angle (WCA) of 0°. Reprinted with permission from [49]. Copyright Wiley 2020.

**Figure 5 nanomaterials-13-02406-f005:**
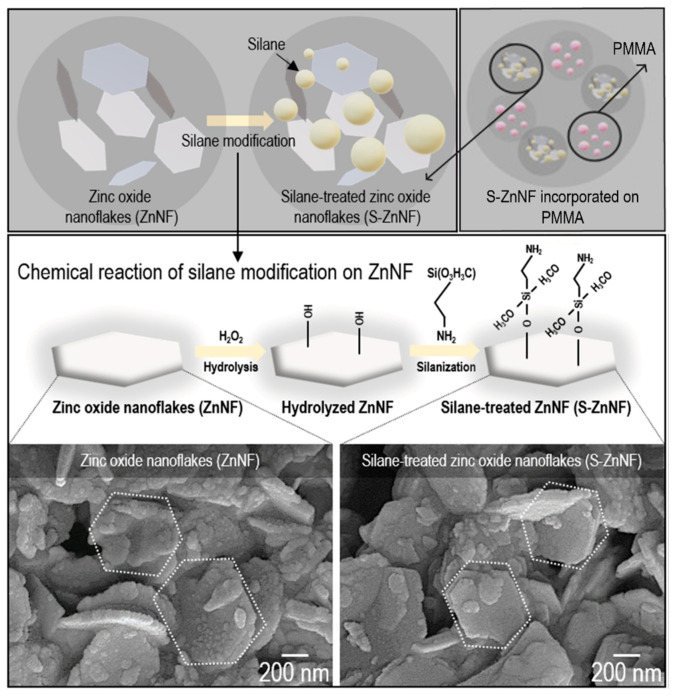
Schematic representation of silane-treated zinc oxide nanoflakes (S-ZnNFs) incorporated into PMMA. Reprinted with permission from [54]. Copyright 2022 Elsevier.

**Figure 6 nanomaterials-13-02406-f006:**
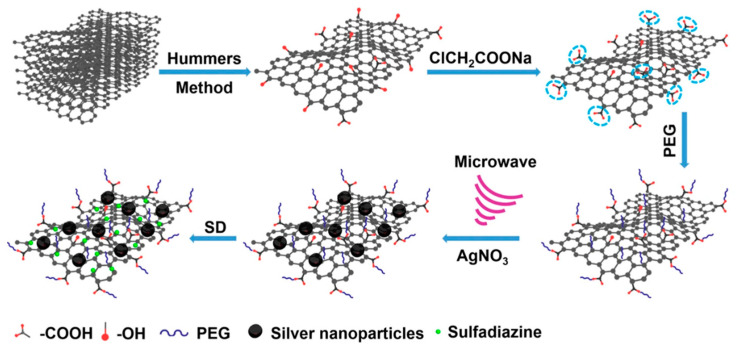
Synthesis scheme of hybrid graphene–AgNP–sulfadiazine nanomaterial [63].

**Figure 7 nanomaterials-13-02406-f007:**
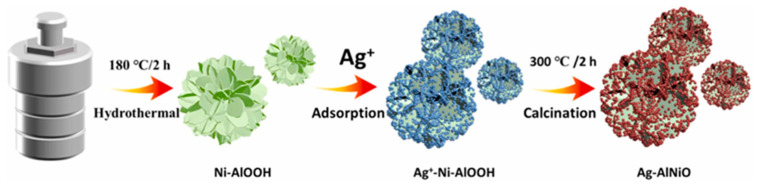
Synthetic description of Ag decoration and assembly on Ni-doped AlOOH. Reprinted with permission from [66]. Copyright 2022 Elsevier.

**Figure 8 nanomaterials-13-02406-f008:**
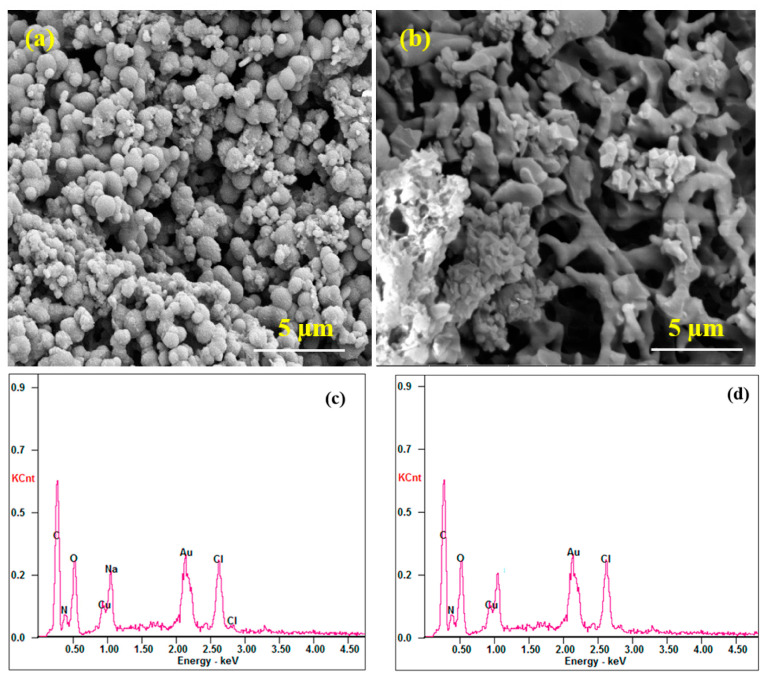
SEM and EDX images of the polymer composites. Notes: (**a**) SEM of polypyrrole (PPY)-Zn@CuO; (**b**) SEM of polyaniline (PANI)-Zn@CuO; (**c**) EDX of PPY-Zn@CuO; (**d**) EDX of PANI-Zn@CuO [67].

**Figure 9 nanomaterials-13-02406-f009:**
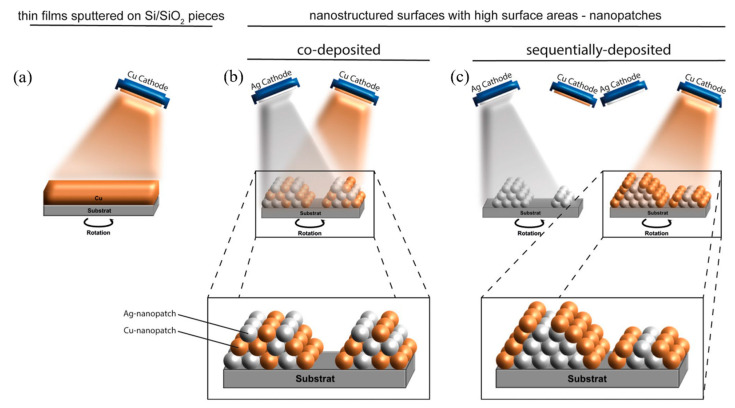
Schematic illustration of the fabrication of Cu and Ag thin-film nanostructures by sputter deposition. (**a**) Elemental Cu sputtered homogeneously; (**b**) Cu and Ag nanopatches co-sputtered; (**c**) Cu and Ag nanopatches sputtered sequentially [16].

**Figure 11 nanomaterials-13-02406-f011:**
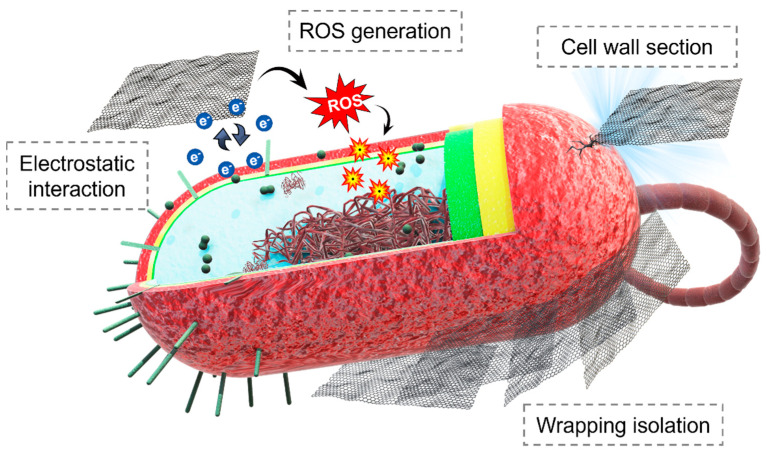
Antimicrobial mechanisms of carbon-based nanomaterials, including the generation of physical damage and oxidative stress through different ways of action: cell wall section, wrapping isolation, ROS generation, or electrostatic interactions.

**Figure 12 nanomaterials-13-02406-f012:**
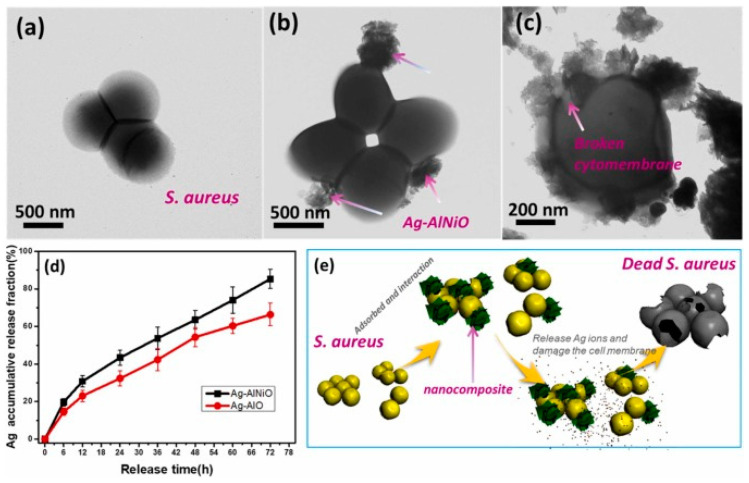
Typical TEM images of *S. aureus* treated by PBS (**a**) and 256 μg/mL of Ag-AlNiO composites (**b**,**c**). (**d**) Ag accumulative release curves of Ag-AlNiO and Ag-AlO. Data represent the mean ± SD (n = 3). (**e**) Description of the killing process of *S. aureus* using Ag-AlNiO nanocomposites. Reprinted with permission from [66]. Copyright 2022 Elsevier.

**Figure 13 nanomaterials-13-02406-f013:**
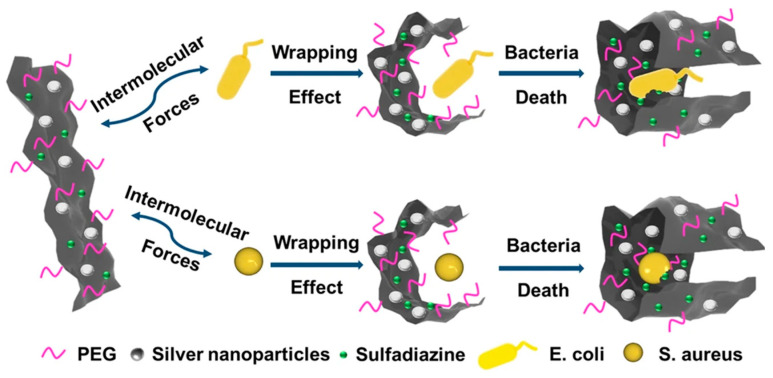
Schematic illustration of the interaction between graphene–AgNPs–sulfadizine and microorganisms [63].

**Figure 14 nanomaterials-13-02406-f014:**
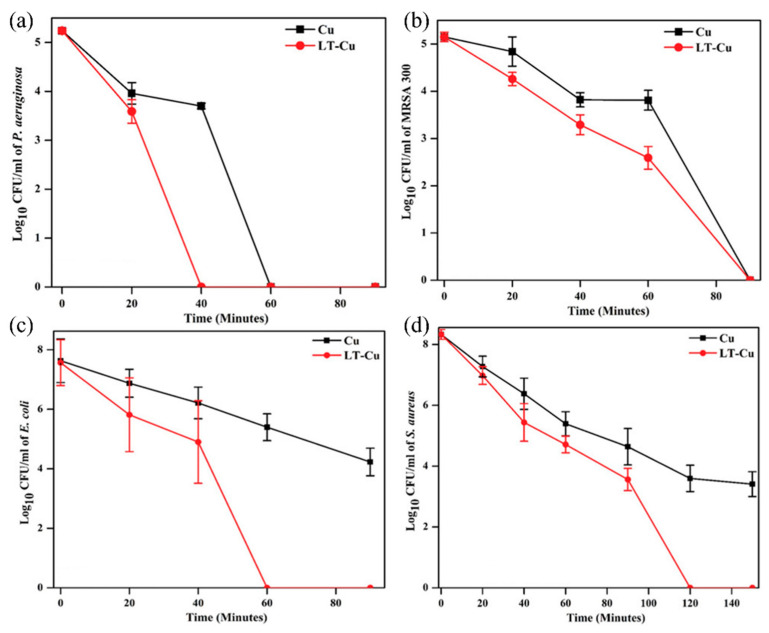
Time profile of bacterial killing by Cu and LT-Cu against (**a**) *P. aeruginosa ATCC 1827*, (**b**) *MRSA USA300*, (**c**) *E. coli ATCC 25 922*, and (**d**) *S. aureus ATCC 25 923* [49].

**Figure 15 nanomaterials-13-02406-f015:**
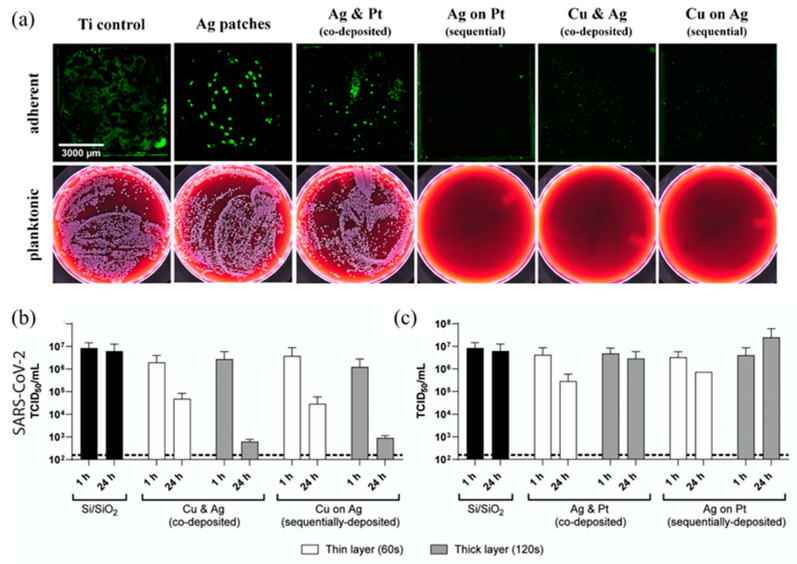
(**a**) Antibacterial activity against *S. aureus* after 24 h of incubation with nanopatches (sputter time: 60 s). Upper figures: Fluorescence images of adherent bacteria. Lower images: representative planktonic bacteria on blood agar plates. (**b**,**c**) Antiviral activity against SARS-CoV-2 for bimetallic nanopatches (Adapted from [16]).

**Figure 16 nanomaterials-13-02406-f016:**
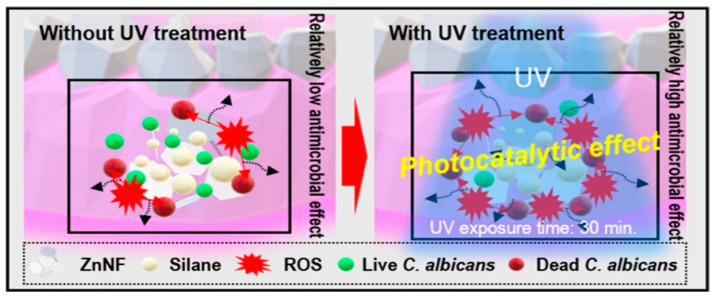
Antimicrobial effect of silane-treated zinc oxide nanoflakes (S-ZnNF) incorporated into PMMA.

**Figure 17 nanomaterials-13-02406-f017:**
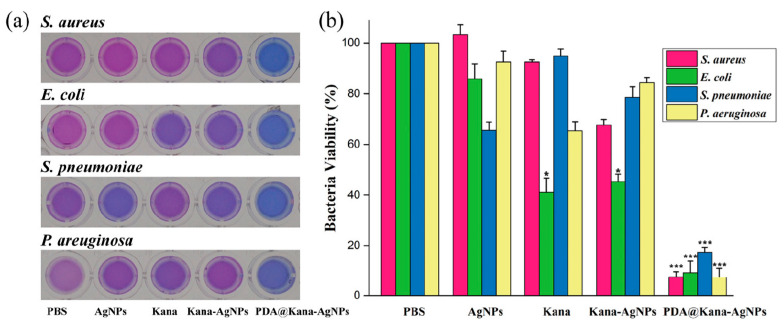
Antimicrobial effect of PDA@Kana-AgNPs and control groups against biofilm-covered bacteria using the resazurin assay. (**a**) The colour change of resazurin reduced by live bacteria in biofilms. (**b**) Relative bacterial viability of the tested bacterial strains. * *p* < 0.05, *** *p* < 0.001. Reprinted with permission from [85]. Copyright 2021 Elsevier.

**Figure 18 nanomaterials-13-02406-f018:**
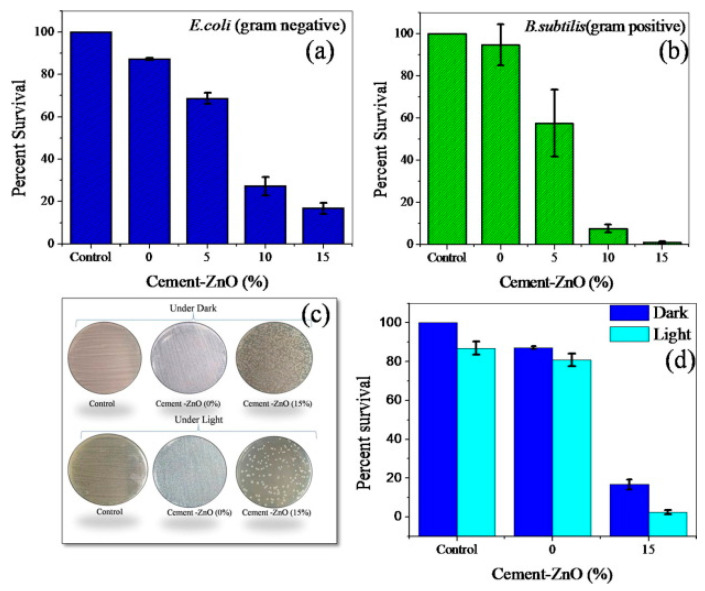
Effect of different concentrations of cement–ZnO composite on bacterial disinfection: (**a**) *E. coli* and (**b**) *B. subtilis.* (**c**) A picture of colony-forming units of *E. coli* on agar plate under dark and sunlight. (**d**) Antibacterial effect of cement–ZnO composite was improved under sunlight. Reprinted with permission from [100]. Copyright 2018 Elsevier.

## Data Availability

The authors confirm that the data supporting the findings of this study are available within the article.

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
