# Peer review of "Recent Advances on the Design and Applications of Antimicrobial Nanomaterials"

_nanomaterials, 2023, doi:10.3390/nano13172406_

Round 1

Reviewer 1 Report

Title: Recent advances on the design and applications of antimicro-2 bial materials

Authors: Clara Ortega-Nieto, Noelia Losada-García, Doina Prodan, Gabriel Furtos  and Jose M. Palomo

Comments:

Google says: “A review article is an article that summarizes the current state of understanding on a topic within a certain discipline.” I believe that is not the case for this manuscript. First, the authors should concentrate on one topic and summarize the current state. It is too much material covered and too little summary.

 I find the manuscript topic interesting and valuable. However, this paper needs much more work in order to meet the technical content criteria.

Some specific comments:

1.      Abstract, line 11: “novel materials”. Those materials are not novel, the methodology may be novel.

2.      Abstract, line 15: “innovative abilities”. Not sure if this is appropriate description.

3.      Intro, line 36: Again “novel and excellent characteristics”

4.      Fig.1 does not give any value to this paper. NPs in general are not the subject of this review. Unless it is significantly improved it should be removed.

5.      Overall introduction is far too brief. This topic deserves much better introduction.

6.      Section 2.1 Synthesis of metal nanoparticles (Ag, Cu, Zn) and metal hybrid nanomaterials. There is unbalanced review of different materials. Ag NP take 1/3 of the page and then there are 3 pages on Cu NPs including substantial part on their own work. Then there is less than one page on ZnO. There is no even mention of GO in their section title. This section needs substantial reorganization. Either parts on Ag and ZnO should be cut out or done properly.

7.      Is the section 2.1 about synthesis or antimicrobial activity?

8.      Section 2.2. “novel applicability” is not clear.

9.      Section 4.1, line: 341: “In addition, the use of nanostructured materials instead of isolated nanoparticles 341 leads to substantial improvements in their properties and characteristics, as has also been 342 seen previously.” Not sure what does that mean, need to be rephrased and supported by references.

10.  Section 4.1. Again there is no real overview, just few examples with far too many details.

11.  Sections 4.2 and 4.3 are getting better than the rest of the manuscript. Though Section 4.3 is too brief.

Author Response

We thank to the referee for helping us to improving our manuscript

All modifications suggested by the referee has been included in the new revised version.

point by point:

  1. Abstract, line 11: “novel materials”. Those materials are not novel, the methodology may be novel. Answer: we are agree and we make the modification
  2. Abstract, line 15: “innovative abilities”. Not sure if this is appropriate description.: answer: we are agree and we made the modification
  3. Intro, line 36: Again “novel and excellent characteristics”. Intro has been changed as referee suggested.
  4. Fig.1 does not give any value to this paper. NPs in general are not the subject of this review. Unless it is significantly improved it should be removed. Answer: we change the fig 1 focus on nanomaterials instead of nanoparticles.
  5. Overall introduction is far too brief. This topic deserves much better introduction. answer:  We are agree and introecution have been extended.
  6. Section 2.1 Synthesis of metal nanoparticles (Ag, Cu, Zn) and metal hybrid nanomaterials. There is unbalanced review of different materials. Ag NP take 1/3 of the page and then there are 3 pages on Cu NPs including substantial part on their own work. Then there is less than one page on ZnO. There is no even mention of GO in their section title. This section needs substantial reorganization. Either parts on Ag and ZnO should be cut out or done properly. Answer:we are agree with the referee. Carbon based hybrid material has been separate in another section 2.2 as referee suggested
  7. Is the section 2.1 about synthesis or antimicrobial activity?
  8. Section 2.2. “novel applicability” is not clear. Answer: It is clearly based on synthetic methods, comments about antimicrobial activity were removed to avoid confusion.
  9. Section 4.1, line: 341: “In addition, the use of nanostructured materials instead of isolated nanoparticles 341 leads to substantial improvements in their properties and characteristics, as has also been 342 seen previously.” Not sure what does that mean, need to be rephrased and supported by references.
  10. Section 4.1. Again there is no real overview, just few examples with far too many details.
  11. Sections 4.2 and 4.3 are getting better than the rest of the manuscript. Though Section 4.3 is too brief

9-11: answer .Phrase has been modified as referee suggested

Reviewer 2 Report

Dear Authors

The review presents a nice review of recent advances in the design and applications of antimicrobial materials. However, there are some suggestions for improvement. 

1) The superhydrophobic coatings were not described in the review.

A.g.

Dual-Functional, Superhydrophobic Coatings with Bacterial Anticontact and Antimicrobial Characteristics

Shuhao Liu, Jeremy Zheng, Li Hao, Yagmur Yegin, Michael Bae, Beril Ulugun, Thomas Matthew Taylor, Ethan A. Scholar, Luis Cisneros-Zevallos, Jun Kyun Oh, and Mustafa Akbulut

ACS Applied Materials & Interfaces 2020 12 (19), 21311-21321

DOI: 10.1021/acsami.9b18928

2) The Utilization of antibacterial nanoparticles in additive manufacturing of advanced materials was also not discussed.  

3) There were presented only well-known basic nanoparticles and not some rear nanoparticles such as cerium oxide: Antibacterial mechanism and activity of cerium oxide nanoparticles. 

To sum up, the review presents a nice summary of basic nanoparticles, but there is a lack of some recent aspects. 

After consideration of the above facts, the manuscript can be accepted. 

The manuscript is structured well. The quality of the images, the number of references and the content paragraphs are well prepared/organised. The language and grammar are of good quality.  

Author Response

 We thank to the referee for helping us to improving our manuscript

All modifications suggested by referee has been included in the new revised version,

Ponit by point:

1) The superhydrophobic coatings were not described in the review.

A.g.

Dual-Functional, Superhydrophobic Coatings with Bacterial Anticontact and Antimicrobial Characteristics

Shuhao Liu, Jeremy Zheng, Li Hao, Yagmur Yegin, Michael Bae, Beril Ulugun, Thomas Matthew Taylor, Ethan A. Scholar, Luis Cisneros-Zevallos, Jun Kyun Oh, and Mustafa Akbulut

ACS Applied Materials & Interfaces 2020 12 (19), 21311-21321

DOI: 10.1021/acsami.9b18928. Answer: we have included new paragraphs about superhydrophobic materials in synthetic and application part as referee suggested including the mentioned references

2) The Utilization of antibacterial nanoparticles in additive manufacturing of advanced materials was also not discussed.  Answer: use of antibacterial nanoparticles as additives is included

3) There were presented only well-known basic nanoparticles and not some rear nanoparticles such as cerium oxide: Antibacterial mechanism and activity of cerium oxide nanoparticles. Answer: additional paragraphs about cerium oxide have been included in the new revised version as referee suggested

Reviewer 3 Report

The submitted manuscript titled "Recent advances on the design and applications of antimicrobial materials" provides a comprehensive overview of recent developments in metal nanoparticles for antimicrobial applications. While it offers valuable insights into this specific area, there are some improvements needed to address the comments below.

1.     The current title suggests a broad focus on antimicrobial materials, yet the content primarily concentrates on nanoparticles. It's essential to provide a general introduction, highlighting the diverse range of antimicrobial materials available, such as polymers (Int J Mol Sci. 2016 Sep; 17(9): 1578. doi: 10.3390/ijms17091578), peptides (Aggregate 2023, 4, e309. Doi: 10.1002/agt2.309), etc. before delving into the specifics of nanoparticles, which is the main focus of the review.

2.     Considering the predominant focus on nanoparticles, the title should be revised to accurately reflect the manuscript's content.

3.     To enhance clarity for readers, the authors should organize the discussion of metal nanoparticles based on their applications, such as medical or energy-related uses. Additionally, incorporating a table summarizing the various types of nanoparticles and their respective applications would make the content more reader-friendly.

Author Response

We thank to the referee for helping us to improving our manuscript

All modifications suggested by referee has been included in the new revised version.

1.     The current title suggests a broad focus on antimicrobial materials, yet the content primarily concentrates on nanoparticles. It's essential to provide a general introduction, highlighting the diverse range of antimicrobial materials available, such as polymers (Int J Mol Sci. 2016 Sep; 17(9): 1578. doi: 10.3390/ijms17091578), peptides (Aggregate 2023, 4, e309. Doi: 10.1002/agt2.309), etc. before delving into the specifics of nanoparticles, which is the main focus of the review. Answer: Introduction have been extended as referee suggested including someof these references 

2.     Considering the predominant focus on nanoparticles, the title should be revised to accurately reflect the manuscript's content. Answer: title have been slightly change following the suggestion fo the referee.

3.     To enhance clarity for readers, the authors should organize the discussion of metal nanoparticles based on their applications, such as medical or energy-related uses. Additionally, incorporating a table summarizing the various types of nanoparticles and their respective applications would make the content more reader-friendly. Answer:  Application part consrning metal nanoparticles have been organized considering the comment of the referee

Round 2

Reviewer 1 Report

The authors have addressed the remarks, though it could have been more substantial change.